# Large-Scale Quadratically Constrained Quadratic Program via Low-Discrepancy Sequences

**Kinjal Basu,  Ankan Saha,  Shaunak Chatterjee**
LinkedIn Corporation
Mountain View, CA 94043
{kbasu, asaha, shchatte}@linkedin.com

## Abstract

We consider the problem of solving a large-scale Quadratically Constrained Quadratic Program. Such problems occur naturally in many scientific and web applications. Although there are efficient methods which tackle this problem, they are mostly not scalable. In this paper, we develop a method that transforms the quadratic constraint into a linear form by sampling a set of low-discrepancy points [16]. The transformed problem can then be solved by applying any state-of-the-art large-scale quadratic programming solvers. We show the convergence of our approximate solution to the true solution as well as some finite sample error bounds. Experimental results are also shown to prove scalability as well as improved quality of approximation in practice.

## 1   Introduction

In this paper we consider the class of problems called quadratically constrained quadratic programming (QCQP) which take the following form:

$$
\begin{aligned}
\underset{\mathbf{x}}{\text{Minimize}} \quad & \frac{1}{2}\mathbf{x}^T\mathbf{P}_0\mathbf{x} + \mathbf{q}_0^T\mathbf{x} + r_0 \\
\text{subject to} \quad & \frac{1}{2}\mathbf{x}^T\mathbf{P}_i\mathbf{x} + \mathbf{q}_i^T\mathbf{x} + r_i \leq 0, \qquad i = 1,\dots,m \\
& \mathbf{A}\mathbf{x} = \mathbf{b},
\end{aligned}
\tag{1}
$$

where $\mathbf{P}_0,\dots,\mathbf{P}_m$ are $n \times n$ matrices. If each of these matrices are positive definite, then the optimization problem is convex. In general, however, solving QCQP is NP-hard, which can be verified by easily reducing a $0-1$ integer programming problem (known to be NP-hard) to a QCQP [4]. In spite of that challenge, they form an important class of optimization problems, since they arise naturally in many engineering, scientific and web applications. Two famous examples of QCQP include the max-cut and boolean optimization [11]. Other examples include alignment of kernels in semi-supervised learning [29], learning the kernel matrix in discriminant analysis [28] as well as more general learning of kernel matrices [21], steering direction estimation for radar detection [15], several applications in signal processing [20], the triangulation in computer vision [3] among others.

Internet applications handling large scale of data, often model trade-offs between key utilities using constrained optimization formulations [1, 2]. When there is independence among the expected utilities (e.g., click, time spent, revenue obtained) of items, the objective or the constraints corresponding to those utilities are linear. However, in most real life scenarios, there is dependence among expected utilities of items presented together on a web page or mobile app. Examples of such dependence are abundant in newsfeeds, search result pages and most lists of recommendations on the internet. If this dependence is expressed through a linear model, it makes the corresponding objective and/or constraint quadratic. This makes the constrained optimization problem a very large scale QCQP, if

the dependence matrix (often enumerated by a very large number of members or updates) is positive definite with co-dependent utilities [6].

Although there are a plethora of such applications, solving this problem on a large scale is still extremely challenging. There are two main relaxation techniques that are used to solve a QCQP, namely, semi-definite programming (SDP) and reformulation-linearization technique (RLT) [11]. However, both of them introduce a new variable $\mathbf{X} = \mathbf{x}\mathbf{x}^T$ so that the problem becomes linear in $\mathbf{X}$. Then they relax the condition $\mathbf{X} = \mathbf{x}\mathbf{x}^T$ by different means. Doing so unfortunately increases the number of variables from $n$ to $O(n^2)$. This makes these methods prohibitively expensive for most large scale applications. There is literature comparing these methods which also provides certain combinations and generalizations[4, 5, 22]. However, they all suffer from the same curse of dealing with $O(n^2)$ variables. Even when the problem is convex, there are techniques such as second order cone programming [23], which can be efficient, but scalability still remains an important issue with prior QCQP solvers.

The focus of this paper is to introduce a novel approximate solution to the convex QCQP problem which can tackle such large-scale situations. We devise an algorithm which approximates the quadratic constraints by a set of linear constraints, thus converting the problem into a quadratic program (QP) [11]. In doing so, we remain with a problem having $n$ variables instead of $O(n^2)$. We then apply efficient QP solvers such as Operator Splitting or ADMM [10, 26] which are well adapted for distributed computing, to get the final solution for problems of much larger scale. We theoretically prove the convergence of our technique to the true solution in the limit. We also provide experiments comparing our algorithm to existing state-of-the-art QCQP solvers to show comparative solutions for smaller data size as well as significant scalability in practice, particularly in the large data regime where existing methods fail to converge. To the best of our knowledge, this technique is new and has not been previously explored in the optimization literature.

**Notation:** Throughout the paper, bold small case letters refer to vectors while bold large-case letters refer to matrices.

The rest of the paper is structured as follows. In Section 2, we describe the approximate problem, important concepts to understand the sampling scheme as well as the approximation algorithm to convert the problem into a QP. Section 3 contains the proof of convergence, followed by the experimental results in Section 4. Finally, we conclude with some discussion in Section 5.

## 2   QCQP to QP Approximation

For sake of simplicity throughout the paper, we deal with a QCQP having a single quadratic constraint. The procedure detailed in this paper can be easily generalized to multiple constraints. Thus, for the rest of the paper, without loss of generality we consider the problem of the form,

$$
\begin{aligned}
\underset{\mathbf{x}}{\text{Minimize}} \quad & (\mathbf{x} - \mathbf{a})^T \mathbf{A}(\mathbf{x} - \mathbf{a}) \\
\text{subject to} \quad & (\mathbf{x} - \mathbf{b})^T \mathbf{B}(\mathbf{x} - \mathbf{b}) \leq \tilde{b}, \\
& \mathbf{C}\mathbf{x} = \mathbf{c}.
\end{aligned}
\tag{2}
$$

This is a special case of the general formulation in (1). For this paper, we restrict our case to $\mathbf{A}$, $\mathbf{B} \in \mathbb{R}^{n \times n}$ being positive definite matrices so that the objective function is strongly convex.

In this section, we describe the linearization technique to convert the quadratic constraint into a set of $N$ linear constraints. The main idea behind this approximation, is the fact that given any convex set in the Euclidean plane, there exists a convex polytope that covers the set. Let us begin by introducing a few notations. Let $\mathcal{P}$ denote the optimization problem (2). Define,

$$
\mathcal{S} := \{\mathbf{x} \in \mathbb{R}^n : (\mathbf{x} - \mathbf{b})^T \mathbf{B}(\mathbf{x} - \mathbf{b}) \leq \tilde{b}\}.
\tag{3}
$$

Let $\partial \mathcal{S}$ denote the boundary of the ellipsoid $\mathcal{S}$. To generate the $N$ linear constraints for this one quadratic constraint, we generate a set of $N$ points, $\mathcal{X}_N = \{\mathbf{x}_1, \ldots, \mathbf{x}_N\}$ such that each $\mathbf{x}_j \in \partial \mathcal{S}$ for $j = 1, \ldots, N$. The sampling technique to select the point set is given in Section 2.1. Corresponding to these $N$ points we get the following set of $N$ linear constraints,

$$
(\mathbf{x} - \mathbf{b})^T \mathbf{B}(\mathbf{x}_j - \mathbf{b}) \leq \tilde{b} \qquad \text{for } j = 1, \ldots, N.
\tag{4}
$$

Looking at it geometrically, it is not hard to see that each of these linear constraints are just tangent planes to $\mathcal{S}$ at $\mathbf{x}_j$ for $j = 1, \ldots, N$. Figure 1 shows a set of six linear constraints for a ellipsoidal

feasible set in two dimensions. Thus, using these $N$ linear constraints we can write the approximate optimization problem, $\mathcal{P}(\mathcal{X}_N)$, as follows.

$$
\begin{aligned}
\underset{\mathbf{x}}{\text{Minimize}} \quad & (\mathbf{x} - \mathbf{a})^T \mathbf{A}(\mathbf{x} - \mathbf{a}) \\
\text{subject to} \quad & (\mathbf{x} - \mathbf{b})^T \mathbf{B}(\mathbf{x}_j - \mathbf{b}) \leq \tilde{b} \qquad \text{for } j = 1, \ldots, N \\
& \mathbf{Cx} = \mathbf{c}.
\end{aligned} \tag{5}
$$

Now instead of solving $\mathcal{P}$, we solve $\mathcal{P}(\mathcal{X}_N)$ for a large enough value of $N$. Note that as we sample more points ($N \to \infty$), our approximation keeps getting better.

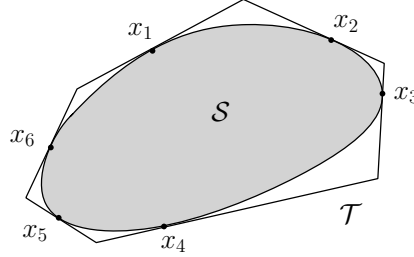

Figure 1: Converting a quadratic constraint into linear constraints. The tangent planes through the 6 points $\mathbf{x}_1, \ldots, \mathbf{x}_6$ create the approximation to $\mathcal{S}$.

## 2.1 Sampling Scheme

The accuracy of the solution of $\mathcal{P}(\mathcal{X}_N)$ solely depends on the choice of $\mathcal{X}_N$. The tangent planes to $\mathcal{S}$ at those $N$ points create a cover of $\mathcal{S}$. We use the notion of a bounded cover, which we define as follows.

*Definition* 1. Let $\mathcal{T}$ be the convex polytope generated by the tangent planes to $\mathcal{S}$ at the points $\mathbf{x}_1, \ldots, \mathbf{x}_N \in \partial \mathcal{S}$. $\mathcal{T}$ is said to be a bounded cover of $\mathcal{S}$ if,

$$
d(\mathcal{T}, \mathcal{S}) := \sup_{\mathbf{t} \in \mathcal{T}} d(\mathbf{t}, \mathcal{S}) < \infty,
$$

where $d(\mathbf{t}, \mathcal{S}) := \inf_{\mathbf{x} \in \mathcal{S}} \|\mathbf{t} - \mathbf{x}\|$ and $\| \cdot \|$ denotes the Euclidean distance.

The first result shows that there exists a bounded cover with only $n + 1$ points.

**Lemma 1.** *Let $\mathcal{S}$ be a $n$ dimensional ellipsoid as defined in* (3)*. Then there exists a bounded cover with $n + 1$ points.*

*Proof.* Note that since $\mathcal{S}$ is a compact convex body in $\mathbb{R}^n$, there exists a location translated version of an $n$-dimensional simplex $T = \{\boldsymbol{x} \in \mathbb{R}_+^n : \sum_{i=1}^n x_i = K\}$ such that $\mathcal{S}$ is contained in the interior of $T$. We can always shrink $T$ such that each edge touches $\mathcal{S}$ tangentially. Since there are $n + 1$ faces, we will get $n + 1$ points whose tangent surface creates a bounded cover. $\qquad \square$

Although Lemma 1 gives a simple constructive proof of a bounded cover, it is not what we are truly interested in. What we want is to construct a bounded cover $\mathcal{T}$ which is as close as possible to $\mathcal{S}$, thus leading to a better approximation. However note that, choosing the points via a naive sampling can lead to arbitrarily bad enlargements of the feasible set and in the worst case might even create a cover which is not bounded. Hence we need an optimal set of points which creates an optimal bounded cover. Formally,

*Definition* 2. $\mathcal{T}^* = \mathcal{T}(\mathbf{x}_1^*, \ldots, \mathbf{x}_N^*)$ is said to be an optimal bounded cover, if

$$
\sup_{\mathbf{t} \in \mathcal{T}^*} d(\mathbf{t}, \mathcal{S}) \leq \sup_{\mathbf{t} \in \mathcal{T}} d(\mathbf{t}, \mathcal{S})
$$

for any bounded cover $\mathcal{T}$ generated by any other $N$-point sets. Moreover, $\{\mathbf{x}_1^*, \ldots, \mathbf{x}_N^*\}$ are defined to be the optimal $N$-point set.

Note that we can think of the optimal $N$-point set as that set of $N$ points which minimize the maximum distance between $\mathcal{T}$ and $\mathcal{S}$, i.e.

$$
\mathcal{T}^* = \underset{\mathcal{T}}{\arg\min} \, d(\mathcal{T}, \mathcal{S}).
$$

It is not hard to see that the optimal $N$-point set on the unit circle in two dimensions are the $N$-th roots of unity, unique up to rotation. This point set also has a very good property. It has been shown that the $N$-th roots of unity minimize the discrete Riesz energy for the unit circle [14, 17]. The concept of Reisz energy also exists in higher dimensions. Thus, generalizing this result, we choose our optimal $N$-point set on $\partial\mathcal{S}$ which tries to minimize the Reisz energy. We briefly describe it below.

### 2.1.1 Riesz Energy

Riesz energy of a point set $A_N = \{\mathbf{x}_1, \ldots, \mathbf{x}_N\}$ is defined as $E_s(A_N) := \sum_{i \neq j=1}^{N} \|\mathbf{x}_i - \mathbf{x}_j\|^{-s}$ for a positive real parameter $s$. There is a vast literature on Riesz energy and its association with "good" configuration of points. It is well known that the measures associated to the optimal point set that minimizes the Riesz energy on $\partial\mathcal{S}$ converge to the normalized surface measure of $\partial\mathcal{S}$ [17]. Thus using this fact, we can associate the optimal $N$-point set to the set of $N$ points that minimize the Riesz energy on $\partial\mathcal{S}$. For more details see [18, 19] and the references therein. To describe these good configurations of points, we introduce the concept of equidistribution. We begin with a "good" or equidistributed point set in the unit hypercube (described in Section 2.1.2) and map it to $\partial\mathcal{S}$ such that the equidistribution property still holds (described in Section 2.1.3).

### 2.1.2 Equidistribution

Informally, a set of points in the unit hypercube is said to be equidistributed, if the expected number of points inside any axis-parallel subregion, matches the true number of points. One such point set in $[0,1]^n$ is called the $(t, m, n)$-net in base $\eta$, which is defined as a set of $N = \eta^m$ points in $[0,1]^n$ such that any axis parallel $\eta$-adic box with volume $\eta^{t-m}$ would contain exactly $\eta^t$ points. Formally, it is a point set that can attain the optimal integration error of $O((\log(N))^{n-1}/N)$ [16] and is usually referred to as a *low-discrepancy* point set. There is vast literature on easy construction of these point sets. For more details on nets we refer to [16, 24].

### 2.1.3 Area preserving map to $\partial\mathcal{S}$

Now once we have a point set on $[0,1]^n$ we try to map it to $\partial\mathcal{S}$ using a measure preserving transformation so that the equidistribution property remains intact. We describe the mapping in two steps. First we map the point set from $[0,1]^n$ to the hyper-sphere $\mathbb{S}^n = \{\mathbf{x} \in \mathbb{R}^{n+1} : \mathbf{x}^T\mathbf{x} = 1\}$. Then we map it to $\partial\mathcal{S}$. The mapping from $[0,1]^n$ to $\mathbb{S}^n$ is based on [12].

The cylindrical coordinates of the $n$-sphere, can be written as

$$\mathbf{x} = \mathbf{x}_n = (\sqrt{1-t_n^2}\mathbf{x}_{n-1}, t_n), \ \ldots \ , \ \mathbf{x}_2 = (\sqrt{1-t_2^2}\mathbf{x}_1, t_2), \ \mathbf{x}_1 = (\cos\phi, \sin\phi)$$

where $0 \leq \phi \leq 2\pi, -1 \leq t_d \leq 1, \mathbf{x}_d \in \mathbb{S}^d$ and $d = (1, \ldots, n)$. Thus, an arbitrary point $\mathbf{x} \in \mathbb{S}^n$ can be represented through angle $\phi$ and heights $t_2, \ldots, t_n$ as,

$$\mathbf{x} = \mathbf{x}(\phi, t_2, \ldots, t_n), \qquad 0 \leq \phi \leq 2\pi, -1 \leq t_2, \ldots, t_n \leq 1.$$

We map a point $\mathbf{y} = (y_1, \ldots, y_n) \in [0,1)^n$ to $\mathbf{x} \in \mathbb{S}^n$ using

$$\varphi_1(y_1) = 2\pi y_1, \qquad \varphi_d(y_d) = 1 - 2y_d \quad (d = 2, \ldots, n)$$

and cylindrical coordinates $\mathbf{x} = \Phi_n(\mathbf{y}) = \mathbf{x}(\varphi_1(y_1), \varphi_2(y_2), \ldots, \varphi_n(y_n))$. The fact that $\Phi_n : [0,1)^n \to \mathbb{S}^n$ is an area preserving map has been proved in [12].

*Remark.* Instead of using $(t, m, n)$-nets and mapping to $\mathbb{S}^n$, we could have also used spherical $t$-designs, the existence of which was proved in [9]. However, construction of such sets is still a tough problem in high dimensions. We refer to [13] for more details.

Finally, we consider the map $\psi$ to translate the point set from $\mathbb{S}^{n-1}$ to $\partial\mathcal{S}$. Specifically we define,

$$\psi(\mathbf{x}) = \sqrt{b}\mathbf{B}^{-1/2}\mathbf{x} + \mathbf{b}. \tag{6}$$

From the definition of $\mathcal{S}$ in (3), it is easy to see that $\psi(\mathbf{x}) \in \partial\mathcal{S}$. The next result shows that this is also an area-preserving map, in the sense of normalized surface measures.

**Lemma 2.** *Let $\psi$ be a mapping from $\mathbb{S}^{n-1} \to \partial\mathcal{S}$ as defined in* (6)*. Then for any set $A \subseteq \partial\mathcal{S}$,*

$$\sigma_n(A) = \lambda_n(\psi^{-1}(A))$$

*where, $\sigma_n, \lambda_n$ are the normalized surface measure of $\partial\mathcal{S}$ and $\mathbb{S}^{n-1}$ respectively.*

*Proof.* Pick any $A \subseteq \partial \mathcal{S}$. Then we can write,

$$\psi^{-1}(A) = \left\{ \frac{1}{\sqrt{\tilde{b}}} \mathbf{B}^{1/2}(\mathbf{x} - \mathbf{b}) : \mathbf{x} \in A \right\}.$$

Now since the linear shift does not change the surface area, we have,

$$\lambda_n(\psi^{-1}(A)) = \lambda_n\left( \left\{ \frac{1}{\sqrt{\tilde{b}}} \mathbf{B}^{1/2}(\mathbf{x} - \mathbf{b}) : \mathbf{x} \in A \right\} \right) = \lambda_n\left( \left\{ \frac{1}{\sqrt{\tilde{b}}} \mathbf{B}^{1/2}\mathbf{x} : \mathbf{x} \in A \right\} \right) = \sigma_n(A),$$

where the last equality follows from the definition of normalized surface measures and noting that $\mathbf{B}^{1/2}\mathbf{x}/\sqrt{\tilde{b}} \in \mathbb{S}^{n-1}$. This completes the proof. $\qquad\square$

Using Lemma 2 we see that the map $\psi \circ \Phi_{n-1} : [0, 1)^{n-1} \to \partial \mathcal{S}$, is a measure preserving map. Using this map and the $(t, m, n-1)$ net in base $\eta$, we derive the optimal $\eta^m$-point set on $\partial \mathcal{S}$. Figure 2 shows how we transform a $(0, 7, 2)$-net in base 2 to a sphere and then to an ellipsoid. For more general geometric constructions we refer to [7, 8].

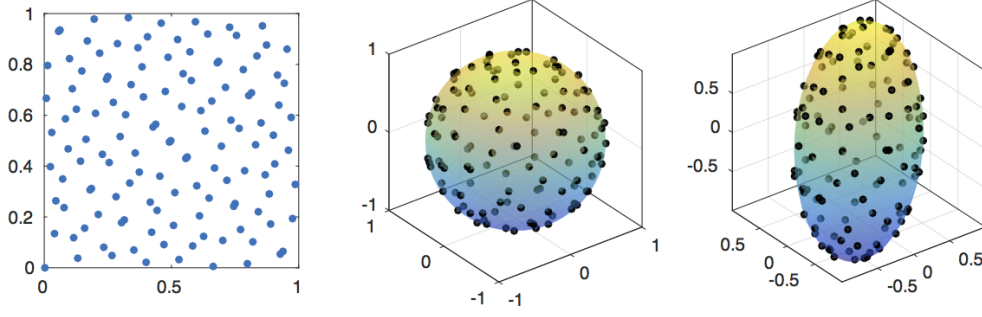

Figure 2: The left panel shows a $(0, 7, 2)$-net in base 2 which is mapped to a sphere in 3 dimensions (middle panel) and then mapped to the ellipsoid as seen in the right panel.

## 2.2 Algorithm and Efficient Solution

From the description in the previous section we are now at a stage to describe the approximation algorithm. We approximate the problem $\mathcal{P}$ by $\mathcal{P}(\mathcal{X}_N)$ using a set of points $\mathbf{x}_1, \dots, \mathbf{x}_N$ as described in Algorithm 1. Once we formulate the problem $\mathcal{P}$ as $\mathcal{P}(\mathcal{X}_N)$, we solve the large scale QP via

---

**Algorithm 1** Point Simulation on $\partial \mathcal{S}$

---

1: Input : $\mathbf{B}, \mathbf{b}, \tilde{b}$ to specify $\mathcal{S}$ and $N = \eta^m$ points
2: Output : $\mathbf{x}_1, \dots, \mathbf{x}_N \in \partial \mathcal{S}$
3: Generate $\mathbf{y}_1, \dots, \mathbf{y}_N$ as a $(t, m, n-1)$-net in base $\eta$.
4: **for** $i \in 1, \dots, N$ **do**
5: $\qquad \mathbf{x}_i = \psi \circ \Phi_{n-1}(\mathbf{y}_i)$
6: **end for**
7: **return** $\mathbf{x}_1, \dots, \mathbf{x}_N$

---

state-of-the-art solvers such as Operator Splitting or Block Splitting approaches [10, 25, 26].

## 3 Convergence of $\mathcal{P}(\mathcal{X}_N)$ to $\mathcal{P}$

In this section, we shall show that if we follow Algorithm 1 to generate the approximate problem $\mathcal{P}(\mathcal{X}_N)$, then we converge to the original problem $\mathcal{P}$ as $N \to \infty$. We shall also prove some finite sample results to give error bounds on the solution to $\mathcal{P}(\mathcal{X}_N)$. We start by introducing some notation.

Let $\mathbf{x}^*, \mathbf{x}^*(N)$ denote the solution to $\mathcal{P}$ and $\mathcal{P}(\mathcal{X}_N)$ respectively and $f(\cdot)$ denote the strongly convex objective function in (2), i.e., for ease of notation

$$f(\mathbf{x}) = (\mathbf{x} - \mathbf{a})^T \mathbf{A}(\mathbf{x} - \mathbf{a}).$$

We begin with our main result.

**Theorem 1.** *Let $\mathcal{P}$ be the QCQP defined in (2) and $\mathcal{P}(\mathcal{X}_N)$ be the approximate QP problem defined in (5) via Algorithm 1. Then, $\mathcal{P}(\mathcal{X}_N) \to \mathcal{P}$ as $N \to \infty$ in the sense that $\lim_{N \to \infty} \|\mathbf{x}^*(N) - \mathbf{x}^*\| = 0$.*

*Proof.* Fix any $N$. Let $\mathcal{T}_N$ denote the optimal bounded cover constructed with $N$ points on $\partial \mathcal{S}$. Note that to prove the result, it is enough to show that $\mathcal{T}_N \to \mathcal{S}$ as $N \to \infty$. This guarantees that linear constraints of $\mathcal{P}(\mathcal{X}_N)$ converge to the quadratic constraint of $\mathcal{P}$, and hence the two problems match. Now since $\mathcal{S} \subseteq \mathcal{T}_N$ for all $N$, it is easy to see that $\mathcal{S} \subseteq \lim_{N \to \infty} \mathcal{T}_N$.

To prove the converse, let $t_0 \in \lim_{N \to \infty} \mathcal{T}_N$ but $t_0 \notin \mathcal{S}$. Thus, $d(t_0, \mathcal{S}) > 0$. Let $t_1$ denote the projection of $t_0$ onto $\mathcal{S}$. Thus, $t_0 \neq t_1 \in \partial \mathcal{S}$. Choose $\epsilon$ to be arbitrarily small and consider any region $A_\epsilon$ around $t_1$ on $\partial \mathcal{S}$ such that $d(x, t_1) \leq \epsilon$ for all $x \in A_\epsilon$. Here $d$ denotes the surface distance function. Now, by the equidistribution property of Algorithm 1 as $N \to \infty$, there exists a point $t^* \in A_\epsilon$, the tangent plane through which cuts the plane joining $t_0$ and $t_1$. Thus, $t_0 \notin \lim_{N \to \infty} \mathcal{T}_N$. Hence, we get a contradiction and the result is proved. $\square$

As a simple Corollary to Theorem 1 it is easy to see that as $\lim_{N \to \infty} |f(\mathbf{x}^*(N)) - f(\mathbf{x}^*)| = 0$. We now move to some finite sample results.

**Theorem 2.** *Let $g : \mathbb{N} \to \mathbb{R}$ such that $\lim_{n \to \infty} g(n) = 0$. Further assume that $\|\mathbf{x}^*(N) - \mathbf{x}^*\| \leq C_1 g(N)$ for some constant $C_1 > 0$. Then, $|f(\mathbf{x}^*(N)) - f(\mathbf{x}^*)| \leq C_2 g(N)$ where $C_2 > 0$ is a constant.*

*Proof.* We begin by bounding the $\|\mathbf{x}^*\|$. Note that since $\mathbf{x}^*$ satisfies the constraint of the optimization problem, we have, $\tilde{b} \geq (\mathbf{x}^* - \mathbf{b})^T \mathbf{B}(\mathbf{x}^* - \mathbf{b}) \geq \sigma_{\min}(\mathbf{B}) \|\mathbf{x}^* - \mathbf{b}\|^2$, where $\sigma_{\min}(\mathbf{B})$ denotes the smallest singular value of $\mathbf{B}$. Thus,

$$\|\mathbf{x}^*\| \leq \|\mathbf{b}\| + \sqrt{\frac{\tilde{b}}{\sigma_{\min}(\mathbf{B})}}. \tag{7}$$

Now, since $f(\mathbf{x}) = (\mathbf{x} - \boldsymbol{a})^T \mathbf{A}(\mathbf{x} - \boldsymbol{a})$ and $\nabla f(\mathbf{x}) = 2\mathbf{A}(\mathbf{x} - \boldsymbol{a})$, we can write

$$f(\mathbf{x}) = f(\mathbf{x}^*) + \int_0^1 \langle \nabla f(\mathbf{x}^* + t(\mathbf{x} - \mathbf{x}^*)), \mathbf{x} - \mathbf{x}^* \rangle dt$$

$$= f(\mathbf{x}^*) + \langle \nabla f(\mathbf{x}^*), \mathbf{x} - \mathbf{x}^* \rangle + \int_0^1 \langle \nabla f(\mathbf{x}^* + t(\mathbf{x} - \mathbf{x}^*)) - \nabla f(\mathbf{x}^*), \mathbf{x} - \mathbf{x}^* \rangle dt$$

$$= I_1 + I_2 + I_3 \text{ (say)}.$$

Now, we can bound the last term as follows. Observe that using Cauchy-Schwarz inequality,

$$|I_3| \leq \int_0^1 |\langle \nabla f(\mathbf{x}^* + t(\mathbf{x} - \mathbf{x}^*)) - \nabla f(\mathbf{x}^*), \mathbf{x} - \mathbf{x}^* \rangle| \, dt$$

$$\leq \int_0^1 \|\nabla f(\mathbf{x}^* + t(\mathbf{x} - \mathbf{x}^*)) - \nabla f(\mathbf{x}^*)\| \|\mathbf{x} - \mathbf{x}^*\| dt$$

$$\leq 2\sigma_{\max}(A) \int_0^1 \|t(\mathbf{x} - \mathbf{x}^*)\| \|\mathbf{x} - \mathbf{x}^*\| dt = \sigma_{\max}(\mathbf{A}) \|\mathbf{x} - \mathbf{x}^*\|^2,$$

where $\sigma_{\max}(\mathbf{A})$ denotes the largest singular value of $\mathbf{A}$. Thus, we have

$$f(\mathbf{x}) = f(\mathbf{x}^*) + \langle \nabla f(\mathbf{x}^*), \mathbf{x} - \mathbf{x}^* \rangle + \tilde{C} \|\mathbf{x} - \mathbf{x}^*\|^2 \tag{8}$$

where $|\tilde{C}| \leq \sigma_{\max}(A)$. Furthermore,

$$|\langle \nabla f(\mathbf{x}^*), \mathbf{x}^*(N) - \mathbf{x}^* \rangle| = |\langle 2A(\mathbf{x}^* - \boldsymbol{a}), \mathbf{x}^*(N) - \mathbf{x}^* \rangle|$$

$$\leq 2\sigma_{\max}(\mathbf{A})(\|\mathbf{x}^*\| + \|\boldsymbol{a}\|) \|\mathbf{x}^*(N) - \mathbf{x}^*\|$$

$$\leq 2C_1 \sigma_{\max}(\mathbf{A}) \left( \sqrt{\frac{\tilde{b}}{\sigma_{\min}(B)}} + \|\mathbf{b}\| + \|\boldsymbol{a}\| \right) g(N), \tag{9}$$

where the last line inequality follows from (7). Combining (8) and (9) the result follows. $\qquad\square$

Note that the function $g$ gives us an idea about how fast $\mathbf{x}^*(N)$ converges $\mathbf{x}^*$. To help, identify the function $g$ we state the following results.

**Lemma 3.** *If $f(\mathbf{x}^*) = f(\mathbf{x}^*(N))$, then $\mathbf{x}^* = \mathbf{x}^*(N)$. Furthermore, if $f(\mathbf{x}^*) \geq f(\mathbf{x}^*(N))$, then $\mathbf{x}^* \in \partial\mathcal{U}$ and $\mathbf{x}^*(N) \notin \mathcal{U}$, where $\mathcal{U} = \mathcal{S} \cap \{\mathbf{x} : \mathbf{Cx} = \mathbf{c}\}$ is the feasible set for (2).*

*Proof.* Let $\mathcal{V} = \mathcal{T}_N \cap \{\mathbf{x} : \mathbf{Cx} = \mathbf{c}\}$. It is easy to see that $\mathcal{U} \subseteq \mathcal{V}$. Assume $f(\mathbf{x}^*) = f(\mathbf{x}^*(N))$, but $\mathbf{x}^* \neq \mathbf{x}^*(N)$. Note that $\mathbf{x}^*, \mathbf{x}^*(N) \in \mathcal{V}$. Since $\mathcal{V}$ is convex, consider a line joining $\mathbf{x}^*$ and $\mathbf{x}^*(N)$. For any point $\lambda_t = t\mathbf{x}^* + (1 - t)\mathbf{x}^*(N)$,

$$f(\lambda_t) \leq tf(\mathbf{x}^*) + (1 - t)f(\mathbf{x}^*(N)) = f(\mathbf{x}^*(N)).$$

Thus, $f$ is constant on the line joining $\mathbf{x}^*$ and $\mathbf{x}^*(N)$. But, it is known that $f$ is strongly convex since $\mathbf{A}$ is positive definite [27]. Thus, there exists only one unique minimum. Hence, we have a contradiction, which proves $\mathbf{x}^* = \mathbf{x}^*(N)$. Now let us assume that $f(\mathbf{x}^*) \geq f(\mathbf{x}^*(N))$. Clearly, $\mathbf{x}^*(N) \notin \mathcal{U}$. Suppose $\mathbf{x}^* \in \mathring{\mathcal{U}}$, the interior of $\mathcal{U}$. Let $\tilde{\mathbf{x}} \in \partial\mathcal{U}$ denote the point on the line joining $\mathbf{x}^*$ and $\mathbf{x}^*(N)$. Clearly, $\tilde{\mathbf{x}} = t\mathbf{x}^* + (1 - t)\mathbf{x}^*(N)$ for some $t > 0$. Thus, $f(\tilde{\mathbf{x}}) < tf(\mathbf{x}^*) + (1 - t)f(\mathbf{x}^*(N)) \leq f(\mathbf{x}^*)$. But $\mathbf{x}^*$ is the minimizer over $\mathcal{U}$. Thus, we have a contradiction, which gives $\mathbf{x}^* \in \partial\mathcal{U}$. This completes the proof. $\qquad\square$

**Lemma 4.** *Following the notation of Lemma 3, if $\mathbf{x}^*(N) \notin \mathcal{U}$, then $\mathbf{x}^*$ lies on $\partial\mathcal{U}$ and no point on the line joining $\mathbf{x}^*$ and $\mathbf{x}^*(N)$ lies in $\mathcal{S}$.*

*Proof.* Since the gradient of $f$ is linear, the result follows from a similar argument to Lemma 3. $\quad\square$

Based on the above two results we can identify the function $g$ by considering the maximum distance of the points lying on the conic cap to the hyperplanes forming it. That is $g(N)$ is the maximum distance between a point $\mathbf{x} \in \partial\mathcal{S}$ and a point in $\mathbf{t} \in \mathcal{T}$ such the line joining $\mathbf{x}$ and $\mathbf{t}$ do not intersect $\mathcal{S}$ and hence, lie completely within the conic section. This is highly dependent on the shape of $\mathcal{S}$ and on the cover $\mathcal{T}_N$. For example, if $\mathcal{S}$ is the unit circle in two dimensions, then the optimal $N$-point set are the $N$-th roots of unity. In which case, there are $N$ equivalent conic sections $\mathcal{C}_1, \ldots, \mathcal{C}_N$ which are created by the intersections of $\partial\mathcal{S}$ with $\mathcal{T}_N$. Figure 3 elaborates these regions.

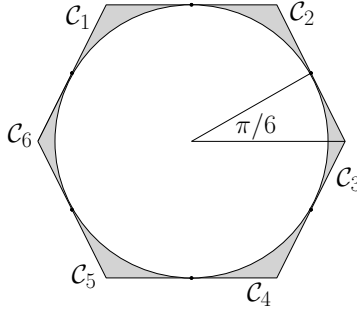

Figure 3: The shaded region shows the 6 equivalent conic regions, $\mathcal{C}_1, \ldots, \mathcal{C}_6$.

To formally define $g(N)$ in this situation, let us define $\mathcal{A}(\mathbf{t}, \mathbf{x})$ to be the set of all points in the line joining $\mathbf{t} \in \mathcal{T}$ and $\mathbf{x} \in \partial\mathcal{S}$. Now, it is easy to see that,

$$g(N) := \max_{i=1,\ldots,N} \sup_{\mathbf{t},\mathbf{x}:\mathcal{A}(\mathbf{t},\mathbf{x})\in\mathcal{C}_i} \|\mathbf{t} - \mathbf{x}\| = \tan\left(\frac{\pi}{N}\right) = O\left(\frac{1}{N}\right), \tag{10}$$

where the bound follows from using the Taylor series expansion of $\tan(x)$. Combining this observation with Theorem 2 shows that in order to get an objective value within $\epsilon$ of the true optimal, we would need $N$ to be a constant multiplier of $\epsilon^{-1}$. More such results can be achieved by such explicit calculations over various different domains $\mathcal{S}$.

## 4 Experimental Results

We compare our proposed technique to the current state-of-the-art solvers of QCQP. Specifically, we compare it to the SDP and RLT relaxation procedures as described in [4]. For small enough problems, we also compare our method to the exact solution by interior point methods. Furthermore, we provide empirical evidence to show that our sampling technique is better than other simpler sampling procedures such as uniform sampling on the unit square or on the unit sphere and then mapping it subsequently to our domain as in Algorithm 1. We begin by considering a very simple QCQP for the form

$$
\begin{aligned}
\underset{\mathbf{x}}{\text{Minimize}} \quad & \mathbf{x}^T \mathbf{A} \mathbf{x} \\
\text{subject to} \quad & (\mathbf{x} - \mathbf{x}_0)^T \mathbf{B}(\mathbf{x} - \mathbf{x}_0) \leq \tilde{b}, \\
& \mathbf{l} \leq \mathbf{x} \leq \mathbf{u}.
\end{aligned} \tag{11}
$$

We randomly sample $\mathbf{A}, \mathbf{B}, \mathbf{x}_0$ and $\tilde{b}$ keeping the problem convex. The lower bound, $\mathbf{l}$ and upper bounds $\mathbf{u}$ are chosen in a way such that they intersect the ellipsoid. We vary the dimension $n$ of the problem and tabulate the final objective value as well as the time taken for the different procedures to converge in Table 1. The stopping criteria throughout our simulation is same as that of Operator Splitting algorithm as presented in [26].

Table 1: The Optimal Objective Value and Convergence Time

| $n$ | Our Method | Sampling on $[0,1]^n$ | Sampling on $\mathbb{S}^n$ | SDP | RLT | Exact |
|---|---|---|---|---|---|---|
| 5 | 3.00 (4.61s) | 2.99 (4.74s) | 2.95 (6. 11s) | 3.07 (0.52s) | 3.08 (0.51s) | 3.07 (0.49) |
| 10 | 206.85 (5.04s) | 205.21 (5.65s) | 206.5 (5.26s) | 252.88 (0.53s) | 252.88 (0.51s) | 252.88 (0.51) |
| 20 | 6291.4 (6.56s) | 4507.8 (6.28s) | 5052.2 (6.69s) | 6841.6 (2.05s) | 6841.6 (1.86s) | 6841.6 (0.54) |
| 50 | 99668 (15.55s) | 15122 (18.98s) | 26239 (17.32s) | $1.11 \times 10^5$ (4.31s) | $1.08 \times 10^5$ (2.96s) | $1.11 \times 10^5$ (0.64) |
| 100 | $1.40 \times 10^6$ (58.41s) | 69746 (1.03m) | $1.24 \times 10^6$ (54.69s) | $1.62 \times 10^6$ (30.41s) | $1.52 \times 10^6$ (15.36s) | $1.62 \times 10^6$ (2.30s) |
| 1000 | $2.24 \times 10^7$ (14.87m) | $8.34 \times 10^6$ (15.63m) | $9.02 \times 10^6$ (15.32m) | NA | NA | NA |
| $10^5$ | $3.10 \times 10^8$ (25.82m) | $7.12 \times 10^7$ (24.59m) | $8.39 \times 10^7$ (27.23m) | NA | NA | NA |
| $10^6$ | $3.91 \times 10^9$ (38.30m) | $2.69 \times 10^8$ (39.15m) | $7.53 \times 10^8$ (37.21m) | NA | NA | NA |

Throughout our simulations, we have chosen $\eta = 2$ and the number of optimal points as $N = \max(1024, 2^m)$, where $m$ is the smallest integer such that $2^m \geq 10n$. Note that even though the SDP and the interior point methods converge very efficiently for small values of $n$, they cannot scale to values of $n \geq 1000$, which is where the strength of our method becomes evident. From Table 1 we observe that the relaxation procedures SDP and RLT fail to converge within an hour of computation time for $n \geq 1000$, whereas all the approximation procedures can easily scale up to $n = 10^6$ variables. Moreover, since the $\mathbf{A}, \mathbf{B}$ were randomly sampled, we have seen that the true optimal solution occurred at the boundary. Therefore, relaxing the constraint to linear forced the solution to occur outside of the feasible set, as seen from the results in Table 1 as well as from Lemma 3. However, that is not a concern, since increasing $N$ will definitely bring us closer to the feasible set. The exact choice of $N$ differs from problem to problem but can be computed as we did with the small example in (10). Finally, the last column in Table 1 is obtained by solving the problem using cvx in MATLAB using via SeDuMi and SDPT3, which gives the true $\mathbf{x}^*$.

Furthermore, our procedure gives the best approximation result when compared to the remaining two sampling schemes. Lemma 3 shows that if the both the objective values are the same then we indeed get the exact solution. To see how much the approximation deviates from the truth, we also plot the log of the relative squared error, i.e. $\log(\|\mathbf{x}^*(N) - \mathbf{x}^*\|^2 / \|\mathbf{x}^*\|^2)$ for each of the sampling

procedures in Figure 4. Throughout this simulation, we keep $N$ fixed at 1024. This is why we see that the error level increases with the increase in dimension. We omit SDP and RLT results in Figure

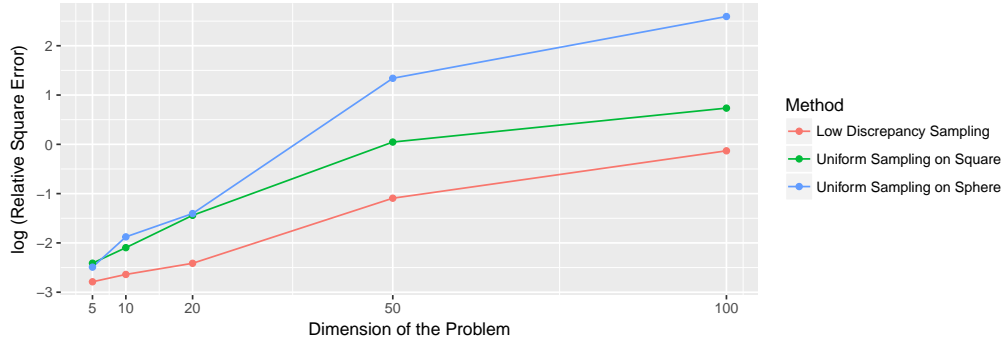

Figure 4: The log of the relative squared error $\log(\|\mathbf{x}^*(N) - \mathbf{x}^*\|^2/\|\mathbf{x}^*\|^2)$ with $N$ fixed at 1024 and varying dimension $n$.

4 since both of them produce a solution very close to the exact minimum for small $n$. If we grow this $N$ with the dimension, then we see that the increasing trend vanishes and we get much more accurate results as seen in Figure 5. We plot both the log of relative squared error as well as the log of the feasibility error, where the feasibility error is defined as

$$\text{Feasibility Error } = \left( (\mathbf{x}^*(N) - \mathbf{x}_0)^T \mathbf{B}(\mathbf{x}^*(N) - \mathbf{x}_0) - \tilde{b} \right)_+$$

where $(\mathbf{x})_+$ denotes the positive part of $\mathbf{x}$.

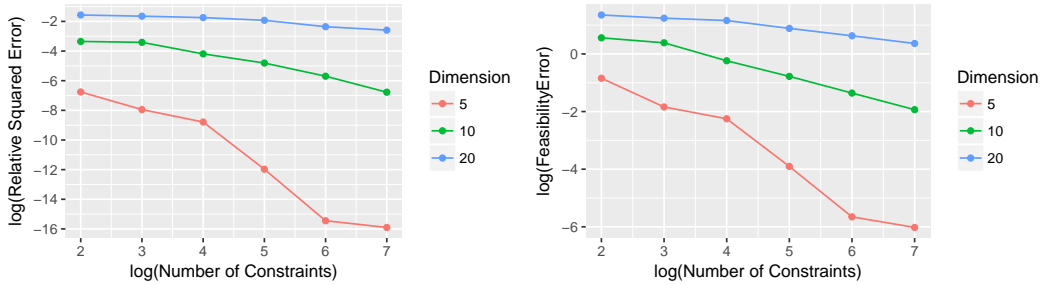

Figure 5: The plot on the left panel and the right panel shows the decay in the relative squared error and the feasibility error respectively, for our method, as we increase $N$ for various dimensions.

From these results, it is clear that our procedure gets the smallest relative error compared to the other sampling schemes, and increasing $N$ brings us closer to the feasible set, with better accurate results.

## 5 Discussion and Future Work

In this paper, we look at the problem of solving a large scale QCQP problem by relaxing the quadratic constraint by a near-optimal sampling scheme. This approximate method can scale up to very large problem sizes, while generating solutions which have good theoretical properties of convergence. Theorem 2 gives us an upper bound as a function of $g(N)$, which can be explicitly calculated for different problems. To get the rate as a function of the dimension $n$, we need to understand how the maximum and minimum eigenvalues of the two matrices $A$ and $B$ grow with $n$. One idea is to use random matrix theory to come up with a probabilistic bound. Because of the nature of complexity of these problems, we believe they deserve special attention and hence we leave them to future work. We also believe that this technique can be immensely important in several applications. Our next step is to do a detailed study where we apply this technique on some of these applications and empirically compare it with other existing large-scale commercial solvers such as CPLEX and ADMM based techniques for SDP.

## Acknowledgment

We would sincerely like to thank the anonymous referees for their helpful comments which has tremendously improved the paper. We would also like to thank Art Owen, Souvik Ghosh, Ya Xu and Bee-Chung Chen for the helpful discussions.

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
