[Reviews · NeurIPS 2017]

Reviewer 1



In this paper, authors develop a method that transforms the quadratic constraint into a linear form by sampling a set of low-discrepancy points. This approach can be used to solve a large-scale quadratically constrained quadratic program by applying any state-of-the-art large-scale quadratic solvers to the transformed problem. The approximation of a quadratic constraint uses a set of linear constraints as the tangent planes through these sampled points. Algorithm 1 presents the approximation algorithm. And also, in section 3, authors show that algorithm 1 generates the approximate problem and it converge to the original problem as N goes to infinity. From the results reported in Table 1, the proposed algorithm can reduce the computation on the size of data larger than 1000. However, it is hard to see the objective difference between the proposed algorithm and the exact solution obtained by SDP. As discussed in the introduction, the targeted optimization problem has been used in many real world applications. In order to evaluate the performance, it might be interesting to see how the approximated solution affects the application results. Another concern is from the results in Table 2. It seems that the relative error is increasing greatly when n increases. In other words, the approximated solution will be very different if n becomes large. The question is: can we trust this algorithm when n is large?

Reviewer 2



Contributions: The paper proposes a sampling based method to approximate QCQPs. The method starts from a sampling of the hypercube, the mapping it to the sphere, and finally to the ellipsoid of interest. Unfortunately the authors don't provide enough evidence why this particular sampling is more efficient than simpler ones. How is the complexity dependence on the dimension? (Net seems to be exponential in dimension). The theorems seem to hide most dimensionality constants. My main question: How does the approach compare experimentally and theoretically to naive sampling? While the main advantage of the contribution is the scalability, the authors do unfortunately not in detail compare (theoretically and practically) the resulting method and its scalability to existing methods for QCQP. The paper is considering convex QCQP only, and does not comment if this problem class is NP hard or not. For the list of applications following this statement, it should be mentioned if those are convex or not, that is if they fall into the scope of the paper or not. This is not the only part of this paper where it occurred to me that the writing lacked a certain amount of rigor and referenced wikipedia style content instead of rigorous definitions. Experiments: No comparison to state-of-the-art second order cone solvers or commercial solvers (e.g. CPLEX and others) is included. Furthermore, as stated now, unfortunately the claimed comparison is not reproducible, as details on the SDP and RLT methods are not included. For example no stopping criteria and implementation details are included. No supplementary material is provided. How was x^* computed in the experimental results? How about adding one real dataset as well from the applications mentioned at the beginning? Riesz energy: "normalized surface measure" either formally define it in the paper, or precisely define it in the appendix, or at least provide a scientific reference for it (not wikipedia). The sentence "We omit details due to lack of space" probably takes as much space as to define it? (Current definition on the sphere is not related to the actual boundary of S). Style: At several points in the paper you mention "For sake of compactness, we do not go deeper into this" and similar. This only makes sense if you at least more precisely state then why you discuss it, or at least make precise what elements you require for your results. In general, if such elements do add to the value of the paper, then they would be provided in an appendix, which the authors here did not. line 105: missing 'the' line 115: in base \eta: Definition makes no sense yet at this point because you didn't specify the roles of m and eta yet (only in the next sentence, for a special case). line 149: space after 'approaches'. remove 'for sake of compactness' == update after rebuttal == I thank the authors for partially clarifying the comparison to other sampling schemes, and on x^*. However, I'm still disappointed by the answer "We are presently working on comparing with large scale commercial solvers such as CPLEX and ADMM based techniques for SDP in several different problems as a future applications paper.". This is not to be left for the future, this comparison to very standard competing methods would be essential to see the merits of the proposed algorithm now, and significantly weakens the current contribution.

Reviewer 3



Review of "Large-Scale Quadratically Constrained Quadratic Program via Low-Discrepancy Sequences." The paper presents a method for solving large-scale QCQPs. The main idea is to approximate a single quadratic constraint by a set of N linear constraints. Section 1 explains the QCQP problem setting with n optimization variables. The two main related methods involve increasing the number of variables to O(n^2), which is intractable for large n. Section 2 explains the linear approximation with equations and Figure 1, which is very helpful for understanding the idea. Section 2.1 introduces the concept of the optimal bounded cover, which is essentially a set of N points/constraints which is the best linear approximation. Section2 2.1.1-2.1.3 provide details about how the N points/constraints are constructed, and Section 2.2 gives pseudocode. Section 3 provides a proof that the linear approximation converges to the solution of the original (quadratically constrained) problem, as the number of points/constraints N is increased. Section 4 provides experimental results on simulated data sets. The paper provides a very clear explanation of an interesting idea with a very elegant geometric interpretation. A strong point of the paper is the figures, which are really do help the reader understand the idea. Another strong point is that there seems to be a few avenues for future work (e.g. other point/constraint sampling strategies). The main weak point of the paper is the experimental section, and the lack of a discussion/analysis about how many constraints N is necessary to achieve a good approximation -- I would hope that it is less than O(n^2). In Table 1 it seems that Our Method gets a lower objective value than the Exact method -- does that mean Our Method is not feasible for the original problem? Is that an issue? Please add some discussion about feasibility. I was a bit disppointed to see the numbers in Table 1 rather than a Figure, which I think would probably be easier for the reader to interpret (and take up less space).